# Anti-Biofilm Effect of Tea Saponin on a *Streptococcus agalactiae* Strain Isolated from Bovine Mastitis

**DOI:** 10.3390/ani10091713

**Published:** 2020-09-22

**Authors:** Fei Shang, Hui Wang, Ting Xue

**Affiliations:** School of Life Sciences, Anhui Agricultural University, Hefei 230036, Anhui, China; shf@ahau.edu.cn (F.S.); wang28hui@163.com (H.W.)

**Keywords:** *Streptococcus agalactiae*, bovine mastitis, biofilm, tea saponin

## Abstract

**Simple Summary:**

Tea saponin (TS), an inexpensive and easily-available plant extract, exhibited antibacterial activity against a *Streptococcus agalactiae* strain isolated from a dairy cow with mastitis. In addition, TS can inhibit the biofilm formation ability of this strain by down-regulating the transcript levels of biofilm-associated genes including *srtA, fbsC, neuA*, and *cpsE*. Hence, TS might be a potential alternative herbal cure for bovine mastitis.

**Abstract:**

*Streptococcus agalactiae* (GBS) is a highly contagious pathogen which not only can cause neonatal meningitis, pneumonia, and septicemia but is also considered to be a major cause of bovine mastitis (BM), leading to large economic losses to the dairy industry worldwide. Like many other pathogenic bacteria, GBS also has the capacity to form a biofilm structure in the host to cause persistent infection. Tea saponin (TS), is one of the main active agents extracted from tea ash powder, and it has good antioxidant and antibacterial activities. In this study, we confirmed that TS has a slight antibacterial activity against a *Streptococcus agalactiae* strain isolated from dairy cow with mastitis and inhibits its biofilm formation. By performing scanning electron microscopy (SEM) experiments, we observed that with addition of TS, the biofilm formed by this GBS strain exhibited looser structure and lower density. In addition, the results of real-time reverse transcription polymerase chain reaction (RT-PCR) experiments showed that TS inhibited biofilm formation by down-regulating the transcription of the biofilm-associated genes including *srtA*, *fbsC*, *neuA*, and *cpsE*.

## 1. Introduction

Bovine mastitis (BM) is the most significant disease of dairy cattle and it can cause substantial decrease in the milk yield and quality, as well as an increase in death rates of cows, bringing about major economic losses in the dairy industry worldwide [1,2,3]. In addition to *Staphylococcus aureus*, *Streptococcus agalactiae*, commonly known as Group B Streptococcus (GBS), is a major causative agent of bovine mastitis [4]. GBS can be colonized in mammary tissue of cows and causes clinical and subclinical mastitis [5]. Transmission of GBS occurs mainly from cow-to-cow via milking equipment, liners, milkers’ hands, or towels in common use [6].

Formation of a biofilm is a common strategy for pathogenic bacteria to adapt to the host environments. Biofilm is defined as a complex structure formed by microbial cells that adhere to media surfaces and are surrounded by a self-produced extracellular polymeric matrix. Under the protection of the extracellular matrix, the microbial cells become resistant to host defenses and tolerant to antibiotic treatment, and usually cause chronic infections which were difficult to treat [7,8,9]. GBS can also form a biofilm-like structure that is usually associated with chronic infections [10,11]. Previous studies have shown that pili play an important role in the biofilm formation of GBS, alongside several other virulence factors including CsrRS, a two component regulatory system and BsaB/FbsC, a protein adhesin which is regulated by the CsrRS system [10,12].

Due to excessive and inappropriate use of antibiotics, drug-resistant bacteria have spread throughout the world. The antimicrobial resistance of mastitis pathogens has drawn much attention around the world [4,13]. For example, in Ethiopia and Estonia, many *S. aureus* and coagulase-negative staphylococcus (CNS) strains have been found to possess penicillin resistance. [13,14]. Studies from India showed that several Gram-negative bacteria exhibited antibiotic resistance to β-lactams and tetracyclines [15]. In addition, as indicated in studies from Canada, antimicrobial resistance genes have been also found in *Streptococcus uberis* and *Streptococcus dysgalactiae,* which were considered to be the important mastitis causative agents [16]. Generally speaking, because of high cost and microbial resistance to the currently-available chemical antibiotics, it is urgently necessary to search for new agents to treat mastitis in cows [17,18].

In recent years, some plant-derived bioactive compounds, which have definite biological functions, have been considered as alternatives to conventional antibiotics [19]. Saponins, a group of glycosides found in many plants, have been confirmed to have anti-inflammatory activity [20,21]. Tea saponin (TS) is a mixture of saponin separated from the seeds, leaves, or roots of the tea tree. It has been reported that TS has relatively high antimicrobial activity against pathogenic dermal fungi and inhibits carrageenan-induced paw oedema in rats [20]. Khan et al. also confirmed that the tea seed saponin mixture they isolated by different methods has antibacterial effects against many Gram-positive and Gram-negative bacteria [22]. However, whether TS has antimicrobial activity against *S. agalactiae* and is associated with bacterial biofilm formation has not been reported. In this study, we explored the antibacterial effect of TS on a GBS strain isolated from bovine mastitis and performed biofilm assays to determine whether TS can affect the biofilm-formation capacity of this strain.

## 2. Materials and Methods

### 2.1. Bacterial Strain and Growth Condition

In this work, the *S. agalactiae* strain GBS2 (hereafter referred as GBS2) was isolated from milk samples in cows with mastitis and was identified by *16S* rDNA sequencing. The cells of GBS2 were cultured in tryptic soy broth (TSB; Oxoid, Basingstoke, UK) medium at 38 °C. The GBS2 serotype was confirmed to be type III, and it was shown to have susceptibilities to norfloxacin, oxacillin, doxycycline, ampicillin, ciprofloxacin, penicillin G, amoxicillin, and ofloxacin and resistance to erythromycin, clindamycin, chloramphenicol, and gentamicin. In addition, this strain possesses several known virulence genes including *fbsA*, *spb1*, *hylB*, *cylE*, and *cspA*.

### 2.2. Inhibitory Effect Assays of TS on GBS2

Growth curves of strain GBS2 were measured as follows: The overnight cultures were diluted to an OD600 of approximately 0.03 in 100 mL of fresh TSB medium without or with different concentrations of TS (Kono Chem. Ltd., Xi’an, China), which was extracted from tea seeds and purified by HPLC. Subsequently, the cultures were incubated at 38 °C for about 24 h with shaking. The OD600 value of each sample was then measured at 2 h intervals by using a UV/Vis spectrophotometer (Thermo Scientific, Pittsburgh, PA, USA).

Colonies of *S. agalactiae* strain GBS2 were transferred into 3 mL of TSB medium and incubated with shaking (180 rpm) for about 16 h at 38 °C. The cultures were transferred into fresh TSB medium and diluted to an optical density of 0.03 (OD_600_ = 0.03), and then the dilutions were dispensed into the 96-well plates (Corning, Steuben, NY, USA) with addition of TS at final concentrations ranging from 0.002 to 2 mg/mL (diluted with sterile water). The bacteria were cultured at 38 °C for about 10–12 h, and then the cultures were 10-fold serial diluted with TSB medium, spread onto the TSB plates, and cultivated at 38 °C for about 16–24 h. After cultivation, viable colony-forming units (CFUs) on every plate were counted, respectively, and the results were compared between the test groups and the control groups. All experiments were repeated at least three times with four parallels.

### 2.3. Biofilm Formation Assays

The experimental method of biofilm assays was according to a previous study [23] and modified as follows: The cells of GBS2 were cultivated in TSB medium for about 16 h and then diluted (1:100 ratio) into fresh TSB (containing 1.0% glucose, 1.0% sodium chloride, and 1.5% milk). The dilutions were immediately transferred into the 96-well plates, TS was added to the cultures at different final concentrations (0.0002 mg/mL, 0.002 mg/mL, 0.02 mg/mL, 0.2 mg/mL, and 2 mg/mL, respectively). Cultures were incubated at 38 °C for about 48 h, and the wells were rinsed five times with water. Subsequently, the plates were stained with 0.5% crystal violet for 15 min, and then rinsed again with water to remove unbound stain. After the plates were dried, the biofilm biomass was measured by using a microplate reader at a wavelength of 560 nm. Every data point was obtained by averaging the absorbance data from at least four replicate wells.

### 2.4. Biofilm Observation by Scanning Electron Microscopy

The biofilm structures of the GBS2 were investigated by SEM XL20 scanning electron microscopy (Philips, Amsterdam, Netherlands). For biofilm formation, the overnight GBS2 cultures were diluted (1:50 ratio) into the fresh TSB broth (containing 1.0% glucose, 1.0% sodium chloride, and 1.5% milk). Sterile coverslips (18 × 18 mm) were placed into the bottom of the wells of the 12-well plates, then the dilutions were transferred onto the coverslips, and the coverslips served as the bacterial attaching surfaces. The cells in the 12-well plates were cultured for about 40–48 h at 38 °C, and then the coverslips were picked out and washed at least two times with PBS buffer. For SEM observation, the samples were prepared according to a previous study [24], Biofilm bacteria were fixed with 5% glutaraldehyde at 4 °C for about 12 h and then dehydrated by using ethanol solution (with serial concentrations: 30%, 50%, 70%, 80%, 95%, and 100%) for at least 20 min at 4 °C. After that, biofilm bacteria with the coverslips were freeze-dried for about 12 h and sputtered onto sample surface of precious metals of about 10 nm thickness. The morphology of GBS2 was observed and photographed at different magnifications.

### 2.5. Isolation and Purification of Total RNA and RT-qPCR Processing

The dilution (1:100 ratio) of the GBS2 cells were transferred into the fresh TSB broth (with or without addition of 2 mg/mL TS), and when the cultures grew to the late exponential phase, bacteria cells were enriched by centrifugation and incubated with Tris-EDTA (TE) buffer (pH 8.0) and 10 g/L lysozyme for 30 min at 37 °C, and then total RNA was extracted by using the Trizol method (Invitrogen, Life Technologies Inc., Carlsbad, CA, USA). DNaseI (TaKaRa, Dalian, China) was used to remove the residual DNA. The PrimeScript 1st Strand cDNA synthesis kit and the SYBR Premix ExTaq (TaKaRa, Dalian, China) was used for RT-qPCR assays, and the RT-qPCR assays were performed by using the StepOne Plus real-time PCR system (Applied Biosystems, Foster City, CA, USA). The 16S cDNA abundance was used to normalize to the quantity of the target genes. All experiments were repeated at least three times with four parallels. The primers used for RT-qPCR assays in this work are listed in Table 1.

### 2.6. Statistical Analysis

The Statistical Product and Service Solutions (SPSS) software (IBM Corp., Armonk, NY, USA) and a one-way ANOVA method were used to analyze the raw data, and the paired *t*-test method was used for statistical comparisons between groups.

## 3. Results

### 3.1. Antibacterial Effect of TS on GBS2

To determine the antibacterial effect of TS on strain GBS2, the growth curves of the cells were measured in the presence of different concentrations of TS. As shown in Figure 1A, there was no significant difference between the growth curves with or without addition of TS, but a slight delay of the logarithmic growth phase. In addition, the CFU assays were also performed to confirm the antibacterial activity of TS against strain GBS2. As shown in Figure 1B, when treated with 2 µg/mL TS, the survival rate of the GBS2 cells exhibited no change compared with that of the control group. However, when the concentration of TS reached 20 µg/mL, the survival rate of GBS2 decreased to about 20% that of the control group, and TS inhibited the survival rate of GBS2 in a dose-dependent manner. According to these data, we suggested that TS has a slight antibacterial activity against strain GBS2 in vitro.

### 3.2. Effects of TS on Biofilm of GBS2

As shown in Figure 2A,B, when 0.0002 mg/mL or 0.002 mg/mL TS was added, no obvious change on biofilm formation of GBS2 was observed; when the concentration of TS reached 0.02 mg/mL, the biofilm formation ability of GBS2 began to be inhibited, and as shown in Figure 2B, the effects of TS on the biomass of biofilm was through a dose-dependent manner. When the concentration of TS reached 2 mg/mL, the biofilm formation of GBS2 was almost completely inhibited.

Additionally, scanning electron microscopy (SEM) experiments were performed to explore the effect of TS on biofilm integrity. As shown in Figure 3, without the addition of TS, GBS2 cells gathered together and formed a thick membrane structure with a relatively complete structure, and a large number of extracellular materials adhered to the cell surface. However, when treated with 2 mg/mL TS, the biomass formed by GBS2 was significantly reduced; the structure of the biofilm was relatively sparse, and the adhesion between the cells was loose. Therefore, we concluded that TS (2 mg/mL) had a significant inhibitory effect on biofilm formation of GBS2.

### 3.3. Effect of TS on Transcriptions of Biofilm-Associated Genes

To investigate the potential mechanism of how TS affects biofilm formation ability of GBS2, RT-qPCR experiments were performed, and the transcript levels of genes associated with biofilm formation in GBS were measured. According to previous studies, several genes (including *srtA*, *fbsC, csrR*, *neuA*, *cpsE,* and *luxS*) have been reported to be involved with the biofilm formation in GBS [10,18,25]. Thus, we measured the transcript levels of these genes when TS was added in GBS2. As shown in Figure 4, when 2 mg/mL TS was added, the transcript levels of *csrR* and *luxS* were not significant changed, but the transcript levels of *srtA*, *fbsC*, *neuA*, and *cpsE* were decreased upon the addition of TS. According to these results, we suggested that TS might inhibit GBS2 biofilm by down-regulating biofilm-associated genes including *srtA*, *fbsC*, *neuA*, and *cpsE*.

## 4. Discussion

Bovine mastitis has been considered to be a great challenge to the dairy industry around the world because it usually results in the reduction of yield and milk quality, and is also associated with the deaths and high treatment costs [19]. *S. agalactiae* was first identified in 1887 as a pathogen causing the mastitis. Although today, GBS has been well recognized as a leading cause of the meningitis in neonates and can cause severe invasive disease in the elderly and in immuno-compromised adults, it is still considered as a major causative agent of bovine mastitis in addition to *S. aureus* [4,26]. Ostensson et al. have reported that the most common pathogen isolated in dairy cows in Southern Vietnam was GBS and a majority of the visited farms were infected with this bacterium [27]. In this study, we found that TS had a slight antibacterial activity against GBS2, a clinical strain isolated from a dairy cow infected with mastitis, and also significantly inhibited the biofilm formation ability of this strain. These results will provide new insights into the treatment of mastitis caused by this bacterium, and might be helpful for reducing the loss of the dairy industries.

Among the common causative agents of bovine mastitis, *S. aureus* is usually proved to be capable of forming biofilms [28]. Although GBS has been demonstrated to be able to form a biofilm-like structure in vitro and in vivo, there are not many reports regarding the association between biofilm formation of GBS and bovine mastitis. However, previous research which was carried out to determine the phenotypic characteristics of GBS isolated from dairy cows infected with mastitis in Iran showed that among 31 GBS isolates, 28 (90.3%) of strains were biofilm producers [29]. In the present study, we also found that the strain we used had strong biofilm formation capacity. Taken together, these data indicated that biofilm formation might also be an important virulence factor which is associated with the pathogenesis of GBS. However, the mechanisms of biofilm formation by GBS still need further exploration.

TS is a natural components of plants which has a series of practical uses. It has been reported that TS has antibacterial effects, and has modified rumen fermentation by reducing the number of rumen protozoa and reducing the loss of intestinal methane [30,31]. Supplementation with TS results in a significant decrease in methane emissions and nitrogen emissions [32]. In addition, TS contributes to improvement in the milk production in lactating dairy cows [33]. However, there was no proof for any association between TS and biofilm formation capacity of bacteria in previous work. Our data revealed that TS can significantly inhibit the biofilm formation capacity of GBS even at a low concentration. In addition, TS also decreased the transcription of several biofilm-associated genes in GBS. Since TS is an inexpensive and easily-available plant extract and possesses significant antibacterial and anti-biofilm activities, it may have a great potential to be developed as a new alternative herbal cure for bovine mastitis.

## 5. Conclusions

This work explored the effect of tea saponin on the growth and biofilm formation in *Streptococcus agalactiae* isolated from dairy cow with mastitis. Results showed that TS had a slight antibacterial activity against GBS2 and inhibited its biofilm formation. We observed that the biofilm of the test group in the presence of TS had a looser structure and lower density compared with the control group without TS. Besides, the results of RT-qPCR indicated that TS inhibited biofilm formation by down-regulating the transcription of the biofilm-associated genes *srtA*, *fbsC*, *neuA*, and *cpsE*.

## Figures and Tables

**Figure 1 animals-10-01713-f001:**
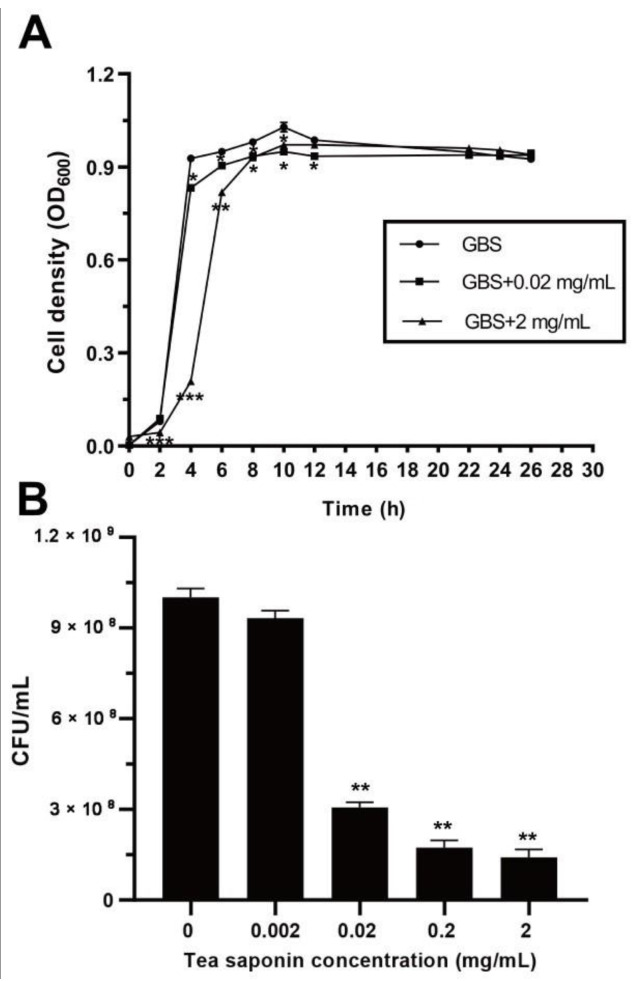
Effects of TS on the growth curve and survival ability of *Streptococcus agalactiae* strain GBS2 (GBS2). (**A**) Growth curves of GBS2 cultured in tryptic soy broth (TSB) medium without or with the corresponding concentration of tea saponin (TS). (**B**) Survival rate assays of GBS2. Colony counts of GBS2 were counted after 12 h of incubation at 38 °C without or with corresponding concentration of TS. The survival rates of the control groups without exposure to TS were designated as 100% (* represents *p* < 0.05 and ** represents *p* < 0.01).

**Figure 2 animals-10-01713-f002:**
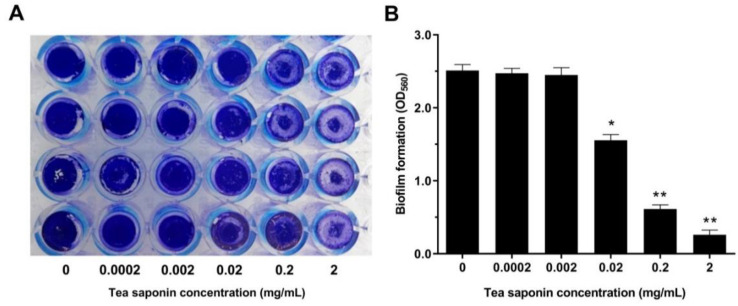
Effects of TS on GBS2 biofilm. (**A**) The crystal-violet stained biofilms of GBS2 in the 96-well plates. (**B**) Results of the biofilm biomass measured by using a microplate reader at a wavelength of 560 nm. Every data point was obtained by averaging the absorbance data from at least four replicate wells (* represents *p* < 0.05 and ** represents *p* < 0.01).

**Figure 3 animals-10-01713-f003:**
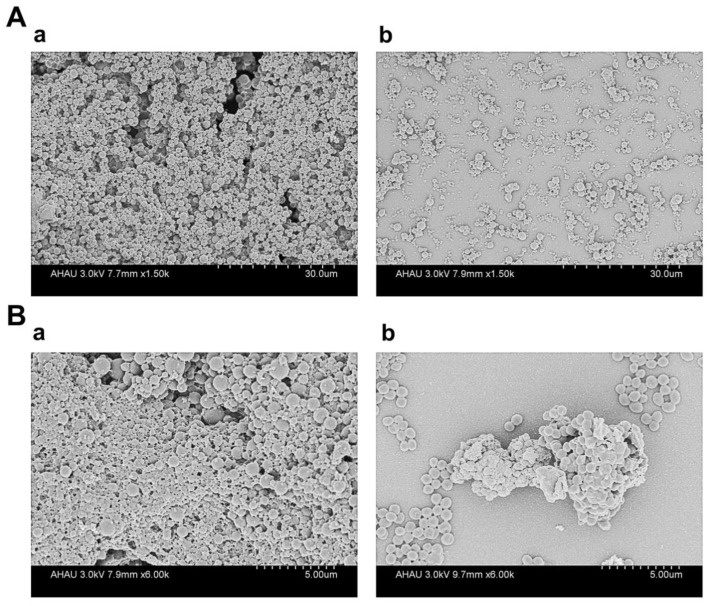
Electron micrographs of GBS2 biofilm monitored by SEM. (**A**) The electron micrographs of GBS2 biofilm at low magnification (1500×) (**a**) without TS and (**b**) with 2 mg/mL TS. (**B**) The electron micrographs of GBS2 biofilm at high magnification (6000×) (**a**) without TS and (**b**) with 2 mg/mL TS.

**Figure 4 animals-10-01713-f004:**
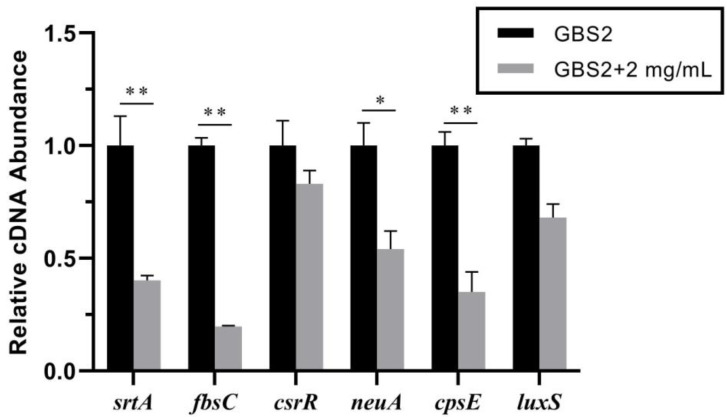
Transcript levels of the biofilm-associated genes determined by RT-qPCR. The transcription of *srtA*, *fbsC, csrR*, *neuA*, *cpsE*, and *luxS* were measured in GBS2 without or with 2 mg/mL TS. The differences of gene expression were calculated by ΔΔCt (Ct = cycle threshold) method using the 16S rDNA gene as the housekeeping gene, normalized by subtracting the Ct value of 16S cDNA from target cDNA. All experiments were repeated at least three times with four parallels (* represents *p* < 0.05 and ** represents *p* < 0.01). Error bars indicate standard deviations.

**Table 1 animals-10-01713-t001:** Oligonucleotide primers used in this work.

Primer Name ^a^	Oligonucleotide (5′-3′)
RT-16s-F	GTAAATGGCGAAGCA
RT-16s-R	TTTGGAAGCGATGAG
RT-cpsE-F	CTTTTACAACGACACGA
RT-cpsE-R	ATCCAAGATACAGACAGC
RT-luxS-F	TCCGCCTTATTCAGC
RT-luxS-R	GACCCCACCAGCAA
RT-neuA-F	ATAAAGGAAGCAATGGA
RT-neuA-R	AGGTGACCGATGACG
RT-csrR-F	CGCTTCGTCTCGTTA
RT-csrR-R	TTCTTTTGTCTTCGTTTC
RT-fbsC-F	TACTCCAAAACCAGTACCACC
RT-fbsC-R	CCTAACATAATCGCTAACCCT
RT-srtA-F	GTGCAGGAACGATGAAGGAA
RT-srtA-R	GGCTCTTGCCAGGTGTATCA

^a^ F = forward; R = reverse.

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
