# Peer review of "Anti-Biofilm Effect of Tea Saponin on a Streptococcus agalactiae Strain Isolated from Bovine Mastitis"

_animals, 2020, doi:10.3390/ani10091713_

Round 1
Reviewer 1 Report
Should be published in Animals after major revision
In their manuscript entitled “The anti-biofilm effect of tea saponin on Streptococcus agalactiae isolated from bovine mastitis”, Fei Shang et al. report the activity of the natural product tea saponin (TS) against a Streptococcus agalactiae strain isolated from a dairy cow with mastitis. The latter is a widespread disease in dairy cattle causing massive economic losses each year. This work is based on previous studies that demonstrated the antimicrobial activity of TS against various bacteria and fungi. Here, TS was evaluated for the first time as regard its potential anti-biofilm effect in S. agalactiae ; a series of results were obtained using various experimental procedures, including: inhibitory effect assays, biofilm formation assays, biofilm examination via SEM, and RT-qPCR expression profiling of biofilm-associated genes.
In my opinion, this paper is quite well-written and pleasant to read, with some adequate references to relevant previous studies. The data reported sounds worthwhile and brings novelty, suggesting that TS could be used as an alternative herbal cure for treating bovine mastitis. However, I think some important information are missing and additional experimental examinations could be done to both ascertain the potential of TS for such application and strengthen the impact of this study. Thus, some improvements are proposed (see below a list of major or minor comments), after which this work would fully deserve to be published in Animals.
Major comments
First, no details are given about the origin and formulation of TS (it is unknown whether it was a commercial product or it was obtained via purification).
All investigations were conducted using a single bacterial strain obtained from a dairy cow with mastitis. More details should be given about this natural isolate (e.g. serotype, antibiotic profile, and virulence genes). The genotype i.e. genetic content in relevant genes (especially those related to biofilm production) should be clearly provided.
Furthermore, a single strain is not enough in my opinion to draw a general conclusion and for this reason the title of this study should be modified for example as follows: “The anti-biofilm effect of tea saponin on *** a *** Streptococcus agalactiae *** strain *** isolated from bovine mastitis”. Along the same line, 'GBS2' should be mentioned rather than only 'GBS' in several instances.
According to the results provided in Figure 1, the kinetic growth of GBS2 in presence of TS is not (at 0.02 mg/mL TS) or only very slightly (at 2 mg/mL TS) delayed (panel A). Consistent with the latter results, no important (thought statistically significant) differences (≤ 1 log10) are found in terms of CFU reduction after incubation for 12 h with up to 2 mg/mL TS (panel B). Thus, I do not agree with the authors when they state for instance on line 220: “Results showed that TS inhibited the growth”. However, TS can exert other antibacterial effect against GBS2 notably through some perturbation of the biofilm formation capacity, which is actually the main assertion of this study.
As for the RT-qPCR assay, other genes related or not related to biofilm formation could have been studied to better examine the specific anti-biofilm effect of TS in GBS2.
Given the target application (i.e. a curative treatment for bovine mastitis), biocompatibility should also be considered (to ascertain the possibility to use such treatment). Cytotoxicity assays would allow to determine whether TS can actually exert an anti-biofilm effect from a dose that does not induce any side-effect for host cells. Author state on line 213: “Our data revealed that TS can significantly inhibit the biofilm formation capacity of GBS even at a low concentration”. However, data reported in Figures 3 and 4 were obtained for 2 mg/mL TS, which is not so low in my view and I wonder whether such concentration would actually be safe for eukaryotic cells.
Other comments
Some references to other relevant recent works should be added, in particular:
- Biofilm production and other virulence factors in Streptococcus spp. isolated from clinical cases of bovine mastitis in Poland. Kaczorek E, Małaczewska J, Wójcik R, Siwicki AK. BMC Vet Res. 2017 Dec 28;13(1):398. doi: 10.1186/s12917-017-1322-y.
- Antimicrobial resistance of biofilm-forming Streptococcus agalactiae isolated from bovine mastitis. Boonyayatra S, Pata P, Nakharuthai P, Chaisri W. J Vet Sci Technol. 2016; 10.4172/2157-7579.1000374.
In Figure 1, error bars are lacking in the graph of panel A. Log10 CFU should be indicated on the y axis in the graph of panel B (instead of “microbial concentration” that sounds not typical and meaningful).
Figure 2 contains some mistakes: Panel A illustrates a picture of a 24-wellplate (not a 96-wellplate as written in the caption of that Figure). In panel B, the text of y axis is not correct (OD492 instead of OD560).
The calculation method used for the treatment of data in Figure 4 is not indicated.
Some sentences must be rephrased/corrected, such as on lines 62-64 : “In this study, we explored the antibacterial effect of TS on GBS2 /// this abbreviation is defined later in the text \\\, and performed biofilm assays to determine whether GBS *** TS *** can affect the biofilm formation capacity of GBS2.”
Author Response
Major comments
Comment: First, no details are given about the origin and formulation of TS (it is unknown whether it was a commercial product or it was obtained via purification).
AU: As you suggested, we have added the origin of TS in the revised manuscript. (Lines 79-80)
Comment: All investigations were conducted using a single bacterial strain obtained from a dairy cow with mastitis. More details should be given about this natural isolate (e.g. serotype, antibiotic profile, and virulence genes). The genotype i.e. genetic content in relevant genes (especially those related to biofilm production) should be clearly provided.
AU: As you suggested, we have added the detailed information about GBS including serotype, antibiotic profile and virulence genes in the revised manuscript. (Lines 71-75)
Comment: Furthermore, a single strain is not enough in my opinion to draw a general conclusion and for this reason the title of this study should be modified for example as follows: “The anti-biofilm effect of tea saponin on *** a *** Streptococcus agalactiae *** strain *** isolated from bovine mastitis”. Along the same line, 'GBS2' should be mentioned rather than only 'GBS' in several instances.
AU: As you suggested, we have changed the title to “Anti-biofilm effect of tea saponin on a Streptococcus agalactiae strain isolated from bovine mastitis”. And we have replaced “GBS” by “GBS2” in several instances.
Comment: According to the results provided in Figure 1, the kinetic growth of GBS2 in presence of TS is not (at 0.02 mg/mL TS) or only very slightly (at 2 mg/mL TS) delayed (panel A). Consistent with the latter results, no important (thought statistically significant) differences (≤ 1 log10) are found in terms of CFU reduction after incubation for 12 h with up to 2 mg/mL TS (panel B). Thus, I do not agree with the authors when they state for instance on line 220: “Results showed that TS inhibited the growth”. However, TS can exert other antibacterial effect against GBS2 notably through some perturbation of the biofilm formation capacity, which is actually the main assertion of this study.
AU: As you suggested, we have corrected the description about the effect of TS on growth of GBS2 in the manuscript. (Lines 137-146)
Comment: As for the RT-qPCR assay, other genes related or not related to biofilm formation could have been studied to better examine the specific anti-biofilm effect of TS in GBS2.
AU: Thanks for your good advice, as shown in Figure 4, we have measured the transcriptions of biofilm-associated genes which have been reported. In addition, pilA and pilB have also been reported to be involved in biofilm formation in GBS, however, in GBS2, these two genes are absent according to our PCR assays.
Comment: Given the target application (i.e. a curative treatment for bovine mastitis), biocompatibility should also be considered (to ascertain the possibility to use such treatment). Cytotoxicity assays would allow to determine whether TS can actually exert an anti-biofilm effect from a dose that does not induce any side-effect for host cells. Author state on line 213: “Our data revealed that TS can significantly inhibit the biofilm formation capacity of GBS even at a low concentration”. However, data reported in Figures 3 and 4 were obtained for 2 mg/mL TS, which is not so low in my view and I wonder whether such concentration would actually be safe for eukaryotic cells.
AU: Several studies have confirmed that TS had cytotoxicity against tumor cells and erythrocyte, in this study, we should pay more attention to the TS cytotoxicity against mammary epithelial cell which have not been reported yet. In our future work, we will focus on this part as you suggested to make the work more complete and solid.
Other comments
Comment: Some references to other relevant recent works should be added, in particular:
- Biofilm production and other virulence factors in Streptococcus spp. isolated from clinical cases of bovine mastitis in Poland. Kaczorek E, Małaczewska J, Wójcik R, Siwicki AK. BMC Vet Res. 2017 Dec 28;13(1):398. doi:10.1186/s12917-017-1322-y.
- Antimicrobial resistance of biofilm-forming Streptococcus agalactiae isolated from bovine mastitis. Boonyayatra S, Pata P, Nakharuthai P, Chaisri W. J Vet Sci Technol. 2016; 10.4172/2157-7579.1000374.
AU: As you suggested, we have cited some references which are related to recent works.
Comment: In Figure 1, error bars are lacking in the graph of panel A. Log10 CFU should be indicated on the y axis in the graph of panel B (instead of “microbial concentration” that sounds not typical and meaningful).
AU: As you suggested, we have added the error bars in the graph of panel A and changed the “microbial concentration” to “CFU” on the y axis in the graph of panel B
Comment: Figure 2 contains some mistakes: Panel A illustrates a picture of a 24-wellplate (not a 96-wellplate as written in the caption of that Figure). In panel B, the text of y axis is not correct (OD492 instead of OD560).
AU: We are sorry for our carelessness, and we have changed “OD492” to “OD560” in the text of y axis. The picture in Panel A is a part of 96-wellplate.
Comment: The calculation method used for the treatment of data in Figure 4 is not indicated.
AU: As your suggestion, we have added the calculation method used for the treatment of data in Figure 4 (Lines 190-193).
Comment: Some sentences must be rephrased/corrected, such as on lines 62-64 : “In this study, we explored the antibacterial effect of TS on GBS2 /// this abbreviation is defined later in the text \\\, and performed biofilm assays to determine whether GBS *** TS *** can affect the biofilm formation capacity of GBS2.”
AU: As your suggestion, we have corrected the sentences on lines 62-64 as follows: In this study, we explored the antibacterial effect of TS on a GBS strain isolated from bovine mastitis, and performed biofilm assays to determine whether TS can affect the biofilm formation capacity of this strain. (Lines 63-66)
Reviewer 2 Report
The manuscript ”The anti-biofilm effect of tea saponin on Streptococcus agalactiae isolated from bovine mastitis” by Fei Shang et al., is poorly written with various grammatical as well as typological errors. It is recommended to revise the manuscript with the help of a native speaker before resubmitting the manuscript.
Scientific part of the manuscript is also very confusing and poorly presented. Please find below some specific comments:
“The anti-biofilm effect of tea saponin on Streptococcus agalactiae isolated from bovine mastitis” – Title can changed to “Anti-biofilm effect of tea saponin on Streptococcus agalactiae isolated from bovine mastitis”.
Line 67: S. agalactiae strain GBS2 change to S. agalactiae strain GBS2 (hereafter referred as GBS2).
How the Tea saponin extracts was prepared…Please mention it briefly…
Line 72: Why 38 not 37? Any specific reason.
Line 73: Write the manufacturer of the instrument (UV/Vis spectrophotometer).
Line 78: What does it mean by concentration 0 mg/mL.
Line 78: what solvent was used to dilute TS.
Figure 1. Authors showed that the TS does not show any inhibitory activity (A except for the delay in Logarithm phase by highest concentration), Whereas at the same time it shows the inhibitory activity (B), when microbial concentration was determined…
Also please write briefly, how the microbial concentration was calculated for Figure 1B (is it determined by CFU?).
Section 2.5: Authors failed to write that what concentration was used for the treatment and eventually for QPCR assay.
Although they mentioned it in the results section that treatment was done with 2mg/mL, still need to be added in methodology section.
The main concern about the whole manuscript is the logic behind biofilm inhibition. Authors need to the contradicting result that if TS is inhibiting microbial concentration from 0.02-2mg/mL, then why it can be proved that the biofilm inhibition observed at the same concentration is not due to the microbial inhibition.
Author Response
Comment: It is recommended to revise the manuscript with the help of a native speaker before resubmitting the manuscript.
AU: As you suggested, the manuscript has been revised by a native speaker.
Comment: “The anti-biofilm effect of tea saponin on Streptococcus agalactiae isolated from bovine mastitis” – Title can changed to “Anti-biofilm effect of tea saponin on Streptococcus agalactiae isolated from bovine mastitis”.
AU: As you suggested, the title has been changed to “Anti-biofilm effect of tea saponin on a Streptococcus agalactiae strain isolated from bovine mastitis”.
Comment: Line 67: S. agalactiae strain GBS2 change to S. agalactiae strain GBS2 (hereafter referred as GBS2).
AU: As you suggested, “S. agalactiae strain GBS2” has been changed to “S. agalactiae strain GBS2 (hereafter referred as GBS2)”. (Line69)
Comment: How the Tea saponin extracts was prepared…Please mention it briefly…
AU: The tea saponin extracts we used in the experiments was purchased from a company. We have added the information in the revised manuscript. (Lines 79-80)
Comment: Line 72: Why 38 not 37? Any specific reason.
AU: Because the average temperature of cow is 38 °C.
Comment: Line 73: Write the manufacturer of the instrument (UV/Vis spectrophotometer).
AU: As you suggested, we have added the manufacturer of the instrument. (Line 82)
Comment: Line 78: What does it mean by concentration 0 mg/mL.
AU: 0 mg/mL means without the addition of TS. To avoid the misunderstanding, we have corrected the sentence. (Lines 86-87)
Comment: Line 78: what solvent was used to dilute TS.
AU: Tea saponin is diluted with sterile water. We have added the related information in the revised manuscript. (Line 87)
Comment: Figure 1. Authors showed that the TS does not show any inhibitory activity (A except for the delay in Logarithm phase by highest concentration), Whereas at the same time it shows the inhibitory activity (B), when microbial concentration was determined… Also please write briefly, how the microbial concentration was calculated for Figure 1B (is it determined by CFU?).
AU: Because the effect of TS on growth of GBS2 mainly exhibited during the logarithmic growth phase, after into the stationary phase, the effect of TS on growth is not obvious. The cells in the CFU assays were cultured not until growing into the stationary phase. Therefore, TS exhibited a slight antibacterial activity against GBS2. We have modified the description in the results. (Lines137-146) In addition, to avoid the misunderstanding, we have changed the “microbial concentration” to “CFU” which were counted in the experiments.
Comment: Section 2.5: Authors failed to write that what concentration was used for the treatment and eventually for QPCR assay.
Although they mentioned it in the results section that treatment was done with 2mg/mL, still need to be added in methodology section.
AU: As you suggested, we have added the concentration of TS used for the treatment in the QPCR assay in methodology section. (Lines 118-119)
Comment: The main concern about the whole manuscript is the logic behind biofilm inhibition. Authors need to the contradicting result that if TS is inhibiting microbial concentration from 0.02-2mg/mL, then why it can be proved that the biofilm inhibition observed at the same concentration is not due to the microbial inhibition.
AU: Because the effect of TS on growth of GBS2 mainly exhibited during the logarithmic growth phase, after into the stationary phase, the effect of TS on growth is not obvious. However, in the biofilm formation assays, the cells were cultured 48h which has been grown into the late stationary phase. Therefore, the biofilm inhibition is not mainly due to the microbial inhibition.
Reviewer 3 Report
This is an interesting paper dealing with the eventual interest of tea saponin in fighting biofilm od Streptococcus agalactiae. The paper may be of interest. However I firmly believe that some modifications should be done before to be considered for publication.
Major comments:
L 66 “Bacterial Strain”
Since it is not a strain coming from a collection, some information about isolation and identification should be added.
L 85-94. The method to measure biofilm formation is relatively weak. There are much more accurate methods, however it is true that many papers are being published by using only Violet Crystal.
- 95-107 The description of the method is probably incomplete, after freeze/drying probably some operation is missing (sputtering?)
Fig 1. This figure 1 is strongly inconsistent. In the curve there is almost no differences between the OD with and without the antimicrobial at 20 h, while in the histogram only 20 % of bacteria are detected. Authors should work on this, or 20 % is a great survival, and antibacterial effect of TS is negligible , or percentage of bacteria means almost noting. In a biofilm 1010 bacteria/µg is a normal value and 20 % of this is 2 x 109 bacteria/g which is a number almost identical to 1010
Fig 3. These are not photographs but electron-micrographs. Bars should be added, it is not enough to write low and high magnification.
Minor comments:
L 75 replace “placed” by “transferred” and “cultivated” by “incubated”4
L 86 replace “bioiflm” by “biofilm”
Author Response
Major comments:
Comment: L 66 “Bacterial Strain”
Since it is not a strain coming from a collection, some information about isolation and identification should be added.
AU: As you suggested, the information about isolation and identification have been added in the revised manuscript. (Lines 69-70)
Comment: L 85-94. The method to measure biofilm formation is relatively weak. There are much more accurate methods, however it is true that many papers are being published by using only Violet Crystal.
AU: Thanks for the careful review. In our previous work, we usually used Violet Crystal staining method to measure biofilm formation. Thus, in this study, we still used this method.
Comment: 95-107 The description of the method is probably incomplete, after freeze/drying probably some operation is missing (sputtering?)
AU: As you suggested, we have complemented the description of the method used for biofilm formation assays. (Lines 114-116)
Comment: Fig 1. This figure 1 is strongly inconsistent. In the curve there is almost no differences between the OD with and without the antimicrobial at 20 h, while in the histogram only 20 % of bacteria are detected. Authors should work on this, or 20 % is a great survival, and antibacterial effect of TS is negligible, or percentage of bacteria means almost noting. In a biofilm 1010 bacteria/µg is a normal value and 20 % of this is 2 x 109 bacteria/g which is a number almost identical to 1010
AU: We agree with your opinion that the antibacterial effect is not significant, and we have modified the description in the results. (Lines 137-146)
Comment: Fig 3. These are not photographs but electron-micrographs. Bars should be added, it is not enough to write low and high magnification.
AU: As you suggested, we have changed “photographs” to “electron-micrographs” in the legend of Figure 3 in the revised manuscript. Bars (5 μM or 30 μM) have been added in the figure.
Minor comments:
Comment: L 75 replace “placed” by “transferred” and “cultivated” by “incubated”4
AU: As you suggested, “placed” has been replaced by “transferred”, and “cultivated” has been replaced by “incubated”. (Line 83)
Comment: L 86 replace “bioiflm” by “biofilm”
AU: As you suggested, “bioiflm” has been replaced by “biofilm”. (Line 100)
Round 2
Reviewer 1 Report
The authors adequately addressed all the points raised.
This manuscript in its current form can be accepted for publication in Animals.
Reviewer 2 Report
The manuscript has improved substantially after careful revision by the authors and hence, i recommend it for publication.
Reviewer 3 Report
The authors eve introduced changes following my first review.
The paperr is now more consistent.